# Proton Treatment Suppresses Exosome Production in Head and Neck Squamous Cell Carcinoma

**DOI:** 10.3390/cancers16051008

**Published:** 2024-02-29

**Authors:** Ameet A. Chimote, Maria A. Lehn, Jay Bhati, Anthony E. Mascia, Mathieu Sertorio, Michael A. Lamba, Dan Ionascu, Alice L. Tang, Scott M. Langevin, Marat V. Khodoun, Trisha M. Wise-Draper, Laura Conforti

**Affiliations:** 1Division of Nephrology, Department of Internal Medicine, University of Cincinnati, Cincinnati, OH 45267, USA; ameet.chimote@uc.edu (A.A.C.); bhatijy@mail.uc.edu (J.B.); 2Division of Hematology-Oncology, Department of Internal Medicine, University of Cincinnati, Cincinnati, OH 45267, USA; schottmi@ucmail.uc.edu (M.A.L.); wiseth@ucmail.uc.edu (T.M.W.-D.); 3Department of Radiation Oncology, University of Cincinnati College of Medicine, Cincinnati, OH 45267, USA; anthony.mascia@cchmc.org (A.E.M.); sertormu@ucmail.uc.edu (M.S.); lambama@ucmail.uc.edu (M.A.L.); ionasctn@ucmail.uc.edu (D.I.); 4Department of Otolarynogology, Head and Neck Surgery, University of Cincinnati College of Medicine, Cincinnati, OH 45267, USA; tangac@ucmail.uc.edu; 5Larner College of Medicine, University of Vermont, Burlington, VT 05405, USA; scott.langevin@uvm.edu; 6University of Vermont Cancer Center, Burlington, VT 05405, USA; 7Division of Rheumatology, Department of Internal Medicine, University of Cincinnati, Cincinnati, OH 45267, USA; marat.khodoun@cchmc.org; 8Division of Immunobiology, Cincinnati Children’s Hospital Medical Center, Cincinnati, OH 45229, USA

**Keywords:** proton therapy, radiation therapy, head and neck cancers, exosomes, photon therapy, cancer immunology, HNSCC, IFN-γ

## Abstract

**Simple Summary:**

In this study, we have explored the impact of a promising new cancer treatment modality called proton therapy (PT) and compared it to traditional X-ray based photon therapy (XRT) for head and neck cancers. Proton therapy is effective because it can be delivered with precision, thus causing fewer side effects. We demonstrate in this study that, when using proton therapy, there was a significant reduction (75%) in the production of small extracellular vesicles, commonly known as exosomes, from cancer cells, which usually suppress the immune system. XRT did not reduce exosome production. Exosomes from both PT and XRT had similar inhibitory effects on immune cells. Our findings suggest that proton therapy might be better at reducing the immune-suppressing effects of cancer exosomes by producing fewer of them.

**Abstract:**

Proton therapy (PT) is emerging as an effective and less toxic alternative to conventional X-ray-based photon therapy (XRT) for patients with advanced head and neck squamous cell carcinomas (HNSCCs) owing to its clustered dose deposition dosimetric characteristics. For optimal efficacy, cancer therapies, including PT, must elicit a robust anti-tumor response by effector and cytotoxic immune cells in the tumor microenvironment (TME). While tumor-derived exosomes contribute to immune cell suppression in the TME, information on the effects of PT on exosomes and anti-tumor immune responses in HNSCC is not known. In this study, we generated primary HNSCC cells from tumors resected from HNSCC patients, irradiated them with 5 Gy PT or XRT, and isolated exosomes from cell culture supernatants. HNSCC cells exposed to PT produced 75% fewer exosomes than XRT- and non-irradiated HNSCC cells. This effect persisted in proton-irradiated cells for up to five days. Furthermore, we observed that exosomes from proton-irradiated cells were identical in morphology and immunosuppressive effects (suppression of IFN-γ release by peripheral blood mononuclear cells) to those of photon-irradiated cells. Our results suggest that PT limits the suppressive effect of exosomes on cancer immune surveillance by reducing the production of exosomes that can inhibit immune cell function.

## 1. Introduction

Head and neck squamous cell carcinoma (HNSCC) is the sixth most prevalent cancer worldwide and among the most immunosuppressive [1,2]. The current standard of care, as per NCCN (National Comprehensive Cancer Network) guidelines, includes surgery followed by photon (X-ray)-based intensity-modulated radiation therapy (IMRT) with or without chemotherapy, definitive chemoradiotherapy (CRT), or induction chemotherapy followed by concurrent CRT, depending on the disease stage and its anatomical location [3,4,5]. However, the delivery of photon therapy (XRT) in HNSCC is complicated because the cancer predominantly presents at a locally advanced stage, and these tumors often lie in close proximity to physiologically critical and radiosensitive organs and structures in the head and neck region [1,3]. XRT is often associated with patient morbidity due to severe acute and long-term toxic effects, including dysphagia, aspiration, mucositis, soft tissue necrosis, and cranial neuropathies [1,3]. Intensity-modulated proton therapy (IMPT) is a promising alternative to conventional photon-based IMRT. Proton therapy (PT) offers advantages over XRT because it can deliver an optimal radiation dose with precision to tumors while sparing the surrounding normal tissues, thus resulting in less treatment-related toxicity with comparable treatment response and disease-free survival [3,6,7]. However, a knowledge gap exists not only in the presumed biological advantage of PT over XRT, but also in the molecular and cellular effects of PT on anti-tumor immune responses [8,9]. Anti-tumor immune responses are of particular interest because of the recent success of immunotherapy in patients with advanced and/or unresectable HNSCC, resulting in full resolution of the disease in some instances, albeit in a small percentage of patients [10]. Furthermore, the American Society for Radiation Oncology (ASTRO) recommends PT for selected HNSCC, such as sinus or skull-based tumors, as the preferred treatment modality, and there are several ongoing clinical trials investigating immunotherapy in conjunction with PT in this patient population [10,11]. Several studies have shown that PT induces a markedly variable and favorable biological response in terms of gene and protein expression in target tissues compared to XRT. PT inhibits factors that contribute to angiogenesis, inflammation, and DNA damage and favors tumor development and growth, while XRT increases these factors [3,6,8,12,13]. Furthermore, transcriptomic studies in murine models of colon cancer have shown that PT activates anti-tumor immune response pathways [9]. However, these findings may be cancer-type-specific, as studies on esophageal cancers have shown that PT and XRT induce similar immune responses [14]. An improved understanding of the effects of PT is necessary to design new treatment combinations for HNSCC and other solid malignancies.

Effective cancer treatment relies on the ability of effector T and natural killer (NK) cells to infiltrate tumors and perform their effector functions, including cytotoxicity and cytokine release, in the tumor microenvironment (TME) [2]. However, these immune functions are suppressed in HNSCC [15,16]. Similar to many solid tumors, HNSCC employs several mechanisms to evade immune surveillance. Multiple elements of the TME, including tumor-derived small extracellular vesicles, also known as exosomes, contribute to immune evasion [15,17,18]. Exosomes are small membrane-encapsulated extracellular vesicles 30–150 nm in diameter. They are released from parent cells (including tumor cells) and allow for intercellular communication via the horizontal transfer of biomolecular cargo, including proteins and nucleic acids [18,19]. Circulating exosomes from HNSCC patients have been shown to suppress the activities of local and peripheral effector T and NK cells, and their plasma levels directly correlate with disease progression and response to therapy [17,18,20,21]. Furthermore, tumor-associated exosomes shape the metastatic niche to support metastatic dissemination by activating stromal cells and suppressing immune surveillance [17]. Overall, there is accumulating evidence of a strong correlation between circulating exosome levels and disease aggressiveness [22,23]. Despite the importance of exosomes in cancer development and progression, there is currently a dearth of information regarding the effects of PT on exosomes in solid malignancies including HNSCC.

In this study, we investigated the effects of PT and XRT on the production of exosomes by primary cancer cells derived from patients with HNSCC. We observed that exosomes from proton-irradiated cells were identical in morphology and immunosuppressive effects on exosomes from photon-irradiated and non-irradiated cells. However, proton irradiation suppressed exosome production by HNSCC cells by 75%. These results suggest that PT may enhance immune surveillance in HNSCC by reducing exosome production.

## 2. Materials and Methods

### 2.1. Human Subjects

Proton and/or photon irradiation studies were conducted on primary cell cultures established from surgically resected tumors obtained from three de-identified HNSCC patients treated at the University of Cincinnati Medical Center. The patients were diagnosed with oral cavity squamous cell tumors that were confirmed by tissue biopsy, were human papilloma virus (HPV) negative, and were not administered any radiation or chemotherapy at the time of tumor resection. Patient characteristics are summarized in Table 1. Informed consent was obtained from all patients. The study and informed consent forms were approved by the University of Cincinnati Institutional Review Board (IRB no. 2014-4755).

### 2.2. Cell Culture

#### 2.2.1. Reagents and Chemicals

Dulbecco’s Modified Eagle Medium (DMEM), RPMI1640, DMEM/F12, phosphate-buffered saline (PBS), penicillin–streptomycin 100X, 0.25% trypsin, 0.05% trypsin, fetal bovine serum (FBS), exosome-depleted FBS, minimum-essential amino acids, sodium pyruvate, L-glutamine, epidermal growth factor (EGF), and insulin–transferrin–selenium (ITS-G) were purchased from Gibco (ThermoFisher Scientific, Waltham, MA, USA); adenine, hydrocortisone, and high cholera toxin from MilliporeSigma (Burlington, MA, USA); and rock inhibitor and primocin from Cayman Chemicals (Ann Arbor, MI, USA) and InvivoGen (San Diego, CA, USA), respectively.

#### 2.2.2. Cal27 Cell Culture

Cal27 cells were obtained from American Tissue Culture Collection (ATCC, Manassas, VA, USA), grown in DMEM with 10% FBS, 1% penicillin–streptomycin, 1% minimum essential amino acids, 6 mM L-glutamine, and 1% sodium pyruvate in the presence of 5% CO_2_ at 37 °C in a humidified incubator.

#### 2.2.3. Generation of Primary HNSCC Patient-Derived Keratinocyte Cell Cultures

Resected tumor tissues were collected in cold Hypo Thermosol FRS preservation solution (MilliporeSigma). The tumors were rinsed with PBS, dissected into 1 × 1 mm fragments, and dissociated with 0.25% trypsin. After trypsin quenching with DMEM containing 10% FBS, pieces of tissue were plated on irradiated (60 Gy) NIH/3T3 feeder cells (ATCC) and left undisturbed in keratinocyte media for 3 days in a humidified 37 °C incubator. The keratinocyte medium contained: DMEM/F12 medium supplemented with 24.2 μg/mL adenine, 1× minimum-essential amino acids, sodium pyruvate, ITS-G, 100 μg/mL primocin, 0.4 μg/mL hydrocortisone, 8.3 ng/mL high cholera toxin, 10 μM of rock inhibitor, 0.05 μg/mL EGF, and 5% FBS. The medium was changed every 2–3 days. Dying irradiated 3T3 feeders and tissue-derived fibroblasts were removed from the culture using 0.05% trypsin and replaced every 4–6 days. Once keratinocytes reached confluence, the irradiated 3T3 feeders and tissue-derived fibroblasts were removed with 0.05% trypsin, followed by the removal of the keratinocytes with 0.25% trypsin, plated onto fresh irradiated 3T3 cells in new dishes, and maintained in culture. In this study, we used primary tumor-derived keratinocytes from three de-identified HNSCC patients: HNC208, HNC285, and HNC365. Both primary HNSCC patient-derived cell cultures and Cal27 cells were tested regularly and were negative for mycoplasma contamination.

### 2.3. Irradiation of HNSCC Cells

HNSCC cells (5 × 10^5^ cells) were seeded on 10 cm tissue culture dishes and maintained in medium supplemented with exosome-depleted FBS. Keratinocytes were plated on culture dishes without 3T3 feeders before irradiation. Cells at 60–70% confluency were irradiated with either PT or XRT, while non-irradiated cells were used as sham controls. For proton irradiation, the plates were sandwiched between two 5.0 cm solid water slabs and placed near the center of a spread-out Bragg peak (SOBP). The irradiation field was 5.0 Gy (physical dose) with an 18 × 18 cm field size and approximately 9.0 cm of SOBP delivered using a pencil beam scanning gantry from a Varian ProBeam delivery nozzle. An absolute dose of 5.0 Gy was calibrated using a NIST-traceable ion chamber and an electrometer using an accepted dose-to-water formulism. The large SOBP minimized any small variation in the water-equivalent depth induced by the plastic tray or wells, as the dose was uniform across the SOBP. For photon irradiation, the plates were placed on 2.0 cm water-equivalent plastic, with additional water-equivalent plastic placed above to provide backscatter. The plastic and cell wells were cantilevered from the end of the treatment table and irradiated from below within a 20 × 20 cm field. This geometry was designed to avoid inhomogeneity effects produced by the air cavity between the radiation source and the irradiated cells at the bottom of the wells and to avoid the dosimetric effects produced by variable attenuation in the treatment table. Then, 5.0 Gy was delivered with 6 MV photons at a dose-rate of 6.0 Gy/min on a Varian TrueBeam LINAC. Treatment monitor units were calculated for the appropriate dose, distance, field size, and depth. Non-irradiated plates were used as controls (sham). For the irradiated and sham plates, the medium was changed immediately after irradiation, and the plates were incubated in a 37 °C incubator in the presence of 5% CO_2_.

### 2.4. Cell Viability

The viability of HNSCC cells was assessed by a trypan blue (MilliporeSigma) exclusion assay.

### 2.5. Exosome Isolation

Exosomes were isolated according to the protocol described by Théry et al. [24]. Briefly, conditioned cell culture medium was collected at 12 h and 5 days from proton-, photon-irradiated, or sham HNSCC cells and centrifuged at 300× *g* for 10 min at 4 °C, followed by 2000× *g* for 20 min at 4 °C to remove dead cells. The supernatant was centrifuged at 10,000× *g* for 30 min at 4 °C to remove the cellular debris. Ultracentrifugation at 100,000× *g* for 70 min in a Beckman Optima LE-80K ultracentrifuge (Beckman Coulter, Indianapolis, IN, USA) at 4 °C enabled exosome sedimentation. The supernatant was discarded, and the exosomal pellet was resuspended in sterile-filtered PBS with a 0.22 μm syringe filter. The ultracentrifugation step was repeated (100,000× *g* for 70 min at 4 °C), the supernatant was discarded, and the exosomes were resuspended in PBS and stored at −80 °C for downstream analysis.

### 2.6. Nanoparticle Tracking Analysis

The amount and size distribution of exosomes were analyzed using a NanoSight NS300 (Malvern, UK) microscope. Exosome preparations (isolated from 5 mL of conditioned medium) were diluted 1:100 to 1:1000 with sterile-filtered PBS with a 0.22 μm syringe filter to achieve 15 to 50 particles per frame for tracking. Each sample was analyzed five times for 30 s each.

### 2.7. Exosome Protein Determination

Exosomes were lysed with cold RIPA buffer (ThermoFisher) and total protein measured in the lysed samples using the Qubit Assay (ThermoFisher) following the manufacturer’s protocol. The amount of measured protein was normalized to the number of viable cells counted in the irradiated plates.

### 2.8. On-Bead Flow Cytometry

An exosomal volume corresponding to 20 μg of protein was used for on-bead flow cytometry using anti-CD63 antibody-coated Dynabeads (ThermoFisher). Briefly, for each sample, 20 μL of the beads was washed with isolation buffer (PBS + 0.1% BSA, from MilliporeSigma) which was sterile filtered with a 0.22 μm syringe filter, mixed with exosome samples, and rocked overnight at 4 °C. The samples were then incubated with PE-anti-human CD63 (RRID: AB_10896786), PE-anti-human CD9 (RRID: AB_2075893), or PE-anti-human CD81 antibodies (RRID: AB_10642024, all antibodies from BioLegend, San Diego, CA, USA). Samples incubated with PE-anti-human IgG (RRID: AB_326435, BioLegend) were used as isotype controls, whereas samples not incubated with any antibodies were used as unstained controls. Samples were analyzed on a BD Fortessa flow cytometer (BD Biosciences, Franklin Lakes, NJ, USA) and the data processed using FlowJo software 10.9.0 (BD Biosciences).

### 2.9. Negative Staining and Transmission Electron Microscope Imaging

Carbon-coated grids (Electron Microscopy Sciences, Hatfield, PA, USA) were glow discharged for 30 s before 3 μL of the exosome sample was absorbed onto the grid surface for 1 min. Excess samples were blotted away before the grids were washed once in a droplet of 2% uranyl acetate (Electron Microscopy Sciences) before a final 1 min soak in 2% uranyl acetate. Excess stain was blotted away, and the grids were allowed to air dry before imaging using a ThermoFisher Scientific Talos L120C 120 kV transmission electron microscope (TEM) equipped with a Ceta 16M CMOS-based detector. Images were acquired using a one-second exposure time with an estimated electron dose rate between 20 and 40 e/A2s and analyzed using ImageJ 1.52a (National Institutes of Health).

### 2.10. PBMC Isolation

Peripheral blood mononuclear cells (PBMCs) were isolated from discarded blood units from healthy individuals’ blood donations the Hoxworth Blood Center (University of Cincinnati) by Ficoll-Paque density gradient centrifugation (Cytiva Life Sciences, Marlborough, MA, USA) [25]. Age information for these donors was not available.

### 2.11. IFN-γ ELISA

Cryopreserved PBMCs were thawed and rested overnight in the presence of 10 ng/mL IL-2 (BioLegend). Then, 1 × 10^6^ cells/mL were treated with 0.13 × 10^9^/mL exosomes isolated from the supernatants of irradiated and non-irradiated HNSCC cells. The exosome-treated PBMCs were then activated for 48 h with plate-bound anti-human CD3 and anti-human CD28 antibodies (BioLegend) [25]. Activated PBMCs that were not incubated with exosomes were used as controls. After 72 h, IFN-γ levels in supernatants were detected using an enzyme-linked immunosorbent assay (ELISA) (Human IFN-γ uncoated ELISA kit, ThermoFisher), according to the manufacturer’s instructions.

### 2.12. Statistical Analysis

Statistical analyses were performed using Student’s *t*-test (paired or unpaired) or analysis of variance (ANOVA). The normality of sample distribution was assessed by the Shapiro–Wilk test, and where the samples failed normality, comparisons were performed by the Mann–Whitney rank sum test or ANOVA on ranks. Post hoc testing on ANOVA was performed by multiple pairwise comparison procedures using either the Holm–Sidak, Tukey, or Dunn’s methods. Statistical analysis was performed using SigmaPlot 13.0 (Grafiti LLC, Palo Alto, CA, USA) and GraphPad Prism 9.0 (GraphPad Software LLC, Boston, MA, USA). *p* ≤ 0.05 was defined as statistically significant. Appropriate statistical tests and corresponding values are described in individual figure legends.

## 3. Results and Discussion

### 3.1. Proton Radiation Decrease Exosome Production by HNSCC Cells

The goal of this study was to investigate the effect of PT and XRT on the exosome production and morphology of HNSCC cells, given that tumor-derived exosomes display the molecular characteristics of their originating tumor cells and modulate anti-tumor immune responses [17,18,19,23]. We cultured primary cells derived from surgically resected tumors from three HNSCC patients (HNC208, HNC285, and HNC365) who were HPV negative and utilized these together with a commercially available HNSCC cell line (Cal27, derived from squamous cell carcinoma of the tongue, HPV negative) to compare the effect of PT and XRT on exosome production. HPV-negative HNSCC tumors are less radiosensitive than HPV-positive tumors [26]. The experimental protocol for measuring exosome production in irradiated cells is shown in Figure 1A. HNSCC cells were viable in proton-irradiated and photon-irradiated plates as well as in non-irradiated (sham) controls (Figure 1B). Small extracellular vesicles were isolated from the cell culture supernatants of irradiated and sham cells by differential ultracentrifugation 12 h after irradiation.

The concentration and size distribution of exosomes were measured using nanoparticle tracking analysis (NTA), and the presence of exosomes was further confirmed by the detection of tetraspanin CD63, CD9, and CD81 (exosome markers) using on-bead flow cytometry and TEM [18,27,28]. For on-bead flow cytometry, the isolated extracellular vesicles were captured on CD63-coated beads, and the abundance of CD63, CD9, and CD81 membrane antigens in the CD63+ exosome subsets was determined by flow cytometry by gating the CD63+ bead population (Figure 1C). As shown in Figure 1D, the extracellular vesicles isolated from the proton- and photon-irradiated and non-irradiated HNSCC cells expressed CD63, CD9, and CD81. We also detected the expression of CD9 and exosome-binding protein TSG101 in the extracellular vesicles isolated from cell culture supernatants of Cal27 cells by Western blotting, thus underscoring the verity of exosome characterization in Figure 1D by on-bead flow cytometry (Appendix A). We visualized the exosomes using TEM to confirm their morphology. As shown in the representative transmission electron micrographs in Figure 1E, the extracellular vesicles for all the treatment groups showed round cup-shaped structures with diameters ranging from 70 to 120 nm, consistent with exosomes.

We then quantified the size distribution and concentration of isolated exosomes by NTA, which showed that compared to non-irradiated and photon-irradiated controls, the 12 h post-irradiation exosome concentration was substantially lower in the supernatant of proton-irradiated HNSCC cells (Figure 2A). To account for the variability in cell numbers across the culture plates, we normalized the exosome concentration measured in the supernatants to the total number of cells counted in each plate. The inhibitory effect of proton radiation on exosome production was consistently observed in the Cal27, HNC208, HNC285, and HNC365 cells. We observed that compared to the XRT and sham controls, exosome concentration was significantly reduced, on average, by ~75% in proton-irradiated HNSCC cells (Figure 2B,C). Similarly, we also observed that PT significantly reduced the amount of exosomal protein (normalized to cell numbers in each irradiated plate) by approximately 45% compared to the sham (Figure 2D). We did not detect any difference in the size distribution of the exosomes isolated from the supernatants of the irradiated cells as compared to sham controls, indicating that irradiation did not affect exosome size (Figure 2E).

Further experiments showed that the inhibition of exosome production by proton in HNSCC cells was not a short-term effect but persisted over time. This suggests long-lasting modifications in the cancer cell mechanisms of exosome production by PT. We measured exosome production in HNSCC cells five days after irradiation. A schematic representation of these experiments is shown in Figure 3A. Briefly, we seeded HNC208, HNC285, and HNC365 cells in Petri dishes and irradiated them with 5 Gy of either protons or photons, whereas non-irradiated cells were used as sham controls (same irradiation regimen as in Figure 1A). The cells were then allowed to grow in exosome-depleted medium for five days, after which, they were frozen. At a later date, we recovered these frozen cells in exosome-depleted medium. We did not observe any reduction in cell viability (Figure 3B). We then isolated exosomes from the conditioned media of these recovered cells and quantified the amount of exosomes produced by the cells over a 12 h period using NTA. The exosome concentration (normalized to cell count) was significantly lower in proton-irradiated cells than in non-irradiated and photon-irradiated cells (Figure 3C). As shown in Figure 3D, the amount of exosomes secreted by HNC208, HNC285, and HNC365 cells following proton irradiation was 55% lower than that secreted by the photon-irradiated and non-irradiated cells (Figure 3E). We did not detect any difference in the size distribution of exosomes from irradiated cells compared to sham controls (Figure 3F).

Overall, these data show that PT suppresses exosome production in HNSCC cells and that proton-induced changes in exosome production persist over time. This effect was unique to PT and was not recapitulated by XRT, suggesting that proton and not photon radiation affects either exosome biogenesis or release in HNSCC cells [18]. It was previously reported that the exposure of a single immortalized HNSCC cell line (BHY) to 3 Gy and 6 Gy XRT increased exosome release compared to non-irradiated cells within 24 h [28]. XRT has also been reported to increase exosome production in prostate and lung cancer. Theodoraki et al. reported an increase in the concentration of circulating tumor exosomes in the plasma of HNSCC patients who received a combination of chemotherapy, immunotherapy, and XRT [20,29,30]. However, in contrast to these reports, we did not observe a significant increase in exosomes released by HNSCC cell lines following XRT as compared to the sham. It is possible that the limited statistical power or different experimental conditions may account for this divergence from the literature regarding the effects of photons on exosome production. Overall, we discovered a significant effect of PT on exosome production that, to the best of our knowledge, has not been reported in the literature for any form of cancer. While our results were consistent among the three irradiated patient-derived cancer cells, these were all from oral cavity cancers. Further studies are warranted in cancer cells from patients with other head and neck cancer subtypes to confirm that our findings extend to multiple forms of head and neck cancers. These findings could have important anti-tumor implications, providing that PT does not exacerbate the immunosuppressive capabilities of tumor-derived exosomes.

### 3.2. Exosomes Derived from Irradiated HNSCC Cells Inhibited IFN-γ Production from PBMCs Irrespective of the Radiation Modality

To study the effect of exosomes produced by proton- or photon-irradiated tumor cells on immune cell functionality, we incubated PBMCs isolated from healthy individuals with equal concentrations of exosomes from proton-, photon-, and sham-irradiated HNSCC cells and measured their ability to secrete IFN-γ. IFN-γ production by activated effector T and NK cells plays an important role in the anti-tumor immune response and is a suitable marker for comparing the anti-tumor immune consequences of exosomes generated by cells treated with different irradiation modalities [31]. To determine the optimal concentration of exosomes needed to produce a biological effect without complete suppression, we first generated a dose–response curve to measure IFN-γ secretion from PBMCs incubated with Cal27-derived exosomes. We observed a 50% inhibition of IFN-γ secretion with an exosome concentration of 0.13 × 10^9^ particles/mL (Figure 4A). We used this concentration to measure IFN-γ secretion in PBMCs incubated with exosomes from sham, proton-, and photon-irradiated HNSCC cells. As shown in Figure 4B, exosomes significantly reduced IFN-γ secretion compared to PBMCs that were not exposed to any exosomes, irrespective of the type of radiation received; this effect was comparable to that induced by exosomes from untreated cells (Figure 4C). These findings suggest that PT does not alter the immunosuppressive capabilities of exosomes. However, PT drastically reduces the number of exosomes released, thereby eliminating or substantially reducing the contribution of tumor-derived exosomes to tumor immune escape, which may ultimately enhance anti-tumor immunity. Consistent with our findings, PT has been reported to activate the interferon signaling transcriptomic signature [9].

## 4. Conclusions

In conclusion, the studies presented here suggest that PT limits the suppressive effect of exosomes on cancer immune surveillance by reducing exosome production, thus blunting an important mechanism of tumor escape. The suppression of exosome production may underscore the benefits of PT over XRT on anti-tumor immunity and the possible advantage of PT in combination with immune checkpoint inhibitors and other forms of immunotherapy.

## Figures and Tables

**Figure 1 cancers-16-01008-f001:**
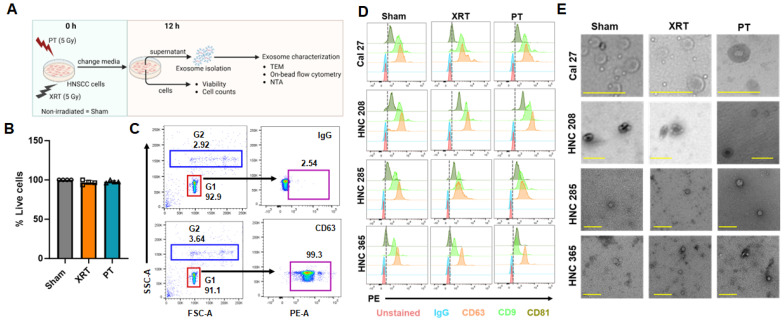
Characterization of exosomes derived from irradiated HNSCC cells. (**A**) Treatment protocol of HNSCC cells and experimental protocol to isolate and characterize exosomes from cells 12 h post irradiation. Three primary HNSCC cell cultures (HNC208, 285, and 365) and an immortalized HNSCC cell line (Cal27) were irradiated with PT or XRT, while non-irradiated cells (Sham) were used as controls. Exosomes were isolated 12 h post irradiation from the cell culture supernatants. (**B**) Viability of HNSCC cell lines 12 h post irradiation was determined by trypan blue exclusion (shown on the y-axis as percent viable cells). Data presented as mean ± SD for *n* = 4 for the 3 HNSCC patient-derived cell cultures and Cal27 cells (each individual represented by a symbol). (**C**) Gating strategy for on-bead flow cytometry experiments to characterize exosomes. Representative gating for exosomes from non-irradiated Cal27 cells captured on anti-CD63 beads is shown here. Briefly, out of the total events acquired (forward vs. side-scatter), we first gated the beads (singlets are gated as G1, doublets are G2). The G1 population was then gated for PE+ signal to determine CD63- (shown here), CD9-, and CD81-expressing exosomes, while PE-IgG antibody was used as isotype control to set the gate. (**D**) Staggered histograms showing CD63, CD9, and CD81 abundance in exosomes isolated from irradiated and sham HNSCC cells that were captured on CD63-antibody-coated beads. Unstained and isotype control (IgG) stained beads were used as negative controls. (**E**) Representative transmission electron micrographs showing exosomes isolated from cell culture supernatants from PT-, XRT-irradiated and sham HNSCC cells (Scale bar = 200 nm).

**Figure 2 cancers-16-01008-f002:**
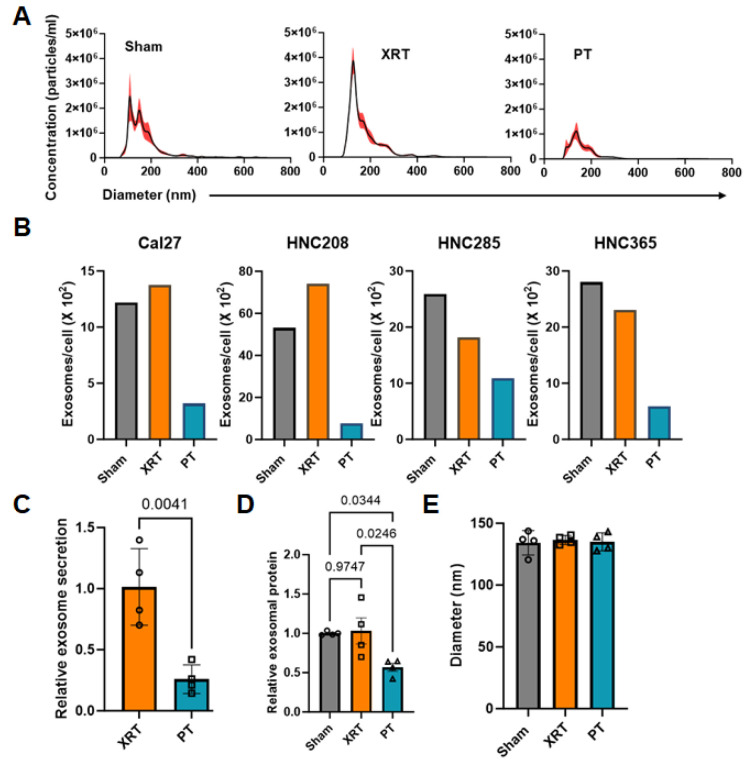
Proton radiations decrease exosome production by HNSCC cells 12 h post irradiation. (**A**) Size distribution and concentration of exosomes isolated from the cell culture supernatant from XRT- or PT-irradiated and non-irradiated (sham) HNSCC cells measured by NTA. A representative analysis of exosomes isolated from irradiated and sham HNC208 cells is shown here. The X-axis represents the diameter of the vesicles, and the Y-axis represents the concentration (particles/mL) of the vesicles. Values are represented as mean ± SEM from 5 independent captures. (**B**) Exosome concentration (normalized to the number of cells in each plate) in the supernatants of XRT- or PT-irradiated and non-irradiated (sham) HNSCC cells (Cal27, HNC208, HNC285 and HNC365). (**C**) Exosomes concentrations shown in (**B**) from each HNSCC cell line were normalized to sham and are presented as relative exosome secretion for XRT- and PT-irradiated cells. Values are presented as mean ± SD for exosomes released from irradiated Cal27, HNC208, HNC285, and HNC365 cells (*n* = 4), each represented by an open circle. Significance was determined by paired *t*-test. (**D**) Total exosomal protein was quantitated in the exosomes isolated from XRT- or PT-irradiated and non-irradiated (sham) HNSCC cell culture supernatants by Qubit Assay and normalized to the number of cells in each plate from which the supernatants were obtained. To determine the changes in the protein levels, exosomal protein levels in sham controls were normalized as 1. Values are presented as mean ± SD for exosomal proteins from irradiated Cal27, HNC208, HNC285, and HNC365 cells (*n* = 4), each represented by an open circle. Significance was determined by one-way ANOVA (*p* = 0.0173), and post hoc testing was carried out using Tukey’s test. (**E**) Mean diameters of exosomes isolated from the supernatants of XRT- or PT-irradiated and non-irradiated (sham) HNSCC cells (*n* = 4). Values are presented as mean ± SD and symbols represent an individual experiment. Significance was determined by one-way ANOVA (*p* = 0.0963).

**Figure 3 cancers-16-01008-f003:**
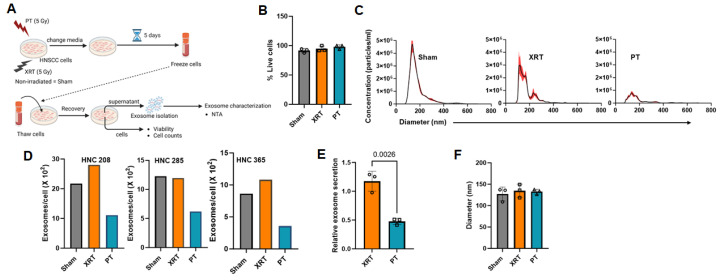
Proton radiations decrease exosome production by HNSCC cells 5 days post irradiation. (**A**) Experimental protocol to isolate and characterize exosomes from irradiated HNSCC cells 5 days post irradiation. Three primary HNSCC patient-derived cell cultures (HNC208, 285, and 365) were irradiated with PT or XRT, while non-irradiated cells (Sham) were used as controls. (**B**) Cell viability of HNSCC cell lines was determined by trypan blue exclusion (shown on the y-axis as percent viable cells). Data presented as mean ± SD for *n* = 3 irradiated patient-derived cells (each individual is represented by the symbol). (**C**) Size distribution and concentration of exosomes isolated from cell culture supernatant from XRT- or PT-irradiated and non-irradiated (sham) HNSCC cells measured by NTA. A representative analysis from irradiated HNC365 primary HNSCC cells is shown here. The X-axis represents the diameter of the vesicles, and the Y-axis represents the concentration (particles/mL) of the vesicles. Values are represented as mean ± SEM from 5 independent captures. (**D**) Exosome concentration in the supernatants normalized to the number of cells in each plate, released from XRT- or PT-irradiated and non-irradiated (sham) HNSCC patient-derived cells (HNC208, HNC285, and HNC365). (**E**) Exosome concentrations shown in (**D**) from each patient-derived cell were normalized to sham and are presented as relative exosome secretion for XRT- and PT-irradiated cells. Values are presented as mean ± SD for exosomes released from irradiated HNC208, HNC285 and HNC365 cells (*n* = 3), where each individual is represented by an open circle. Significance was determined by paired *t*-test. (**F**) Mean diameters of exosomes isolated from the supernatants of XRT or PT- irradiated and non-irradiated (sham) HNSCC cells (*n* = 3). Values are presented as mean ± SD and symbols represent an individual. Significance was determined by one-way ANOVA (*p* = 0.7886).

**Figure 4 cancers-16-01008-f004:**
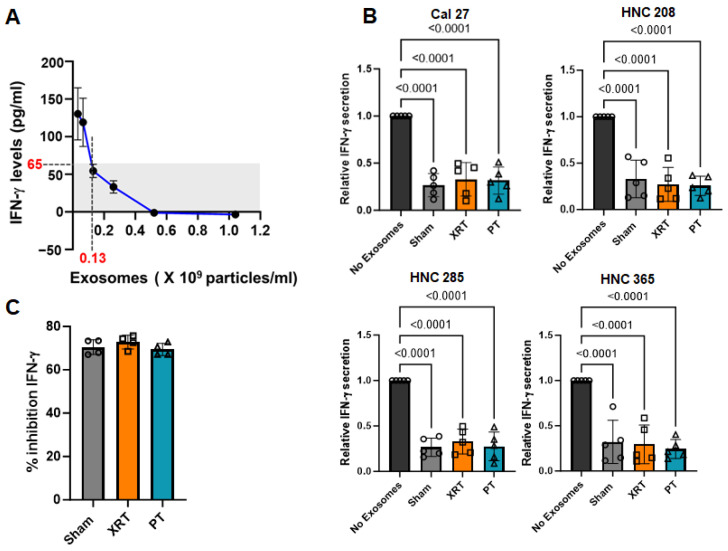
PBMCs from healthy donors treated with exosomes derived from irradiated HNSCC cells show decreased IFN-γ production. (**A**) IFN-γ levels (pg/mL) in HD (healthy donor) PBMCs treated with different concentrations of exosomes isolated from supernatants of non-irradiated Cal27 cells and activated with TG. Values are presented as mean ± SD for IFN-γ levels measured in *n* = 3 HD PBMCs treated with exosomes. Also, the extrapolated concentration of exosomes (0.13 × 10^9^ particles/mL, dotted line) at which we measured 50% IFN-γ levels (65 pg/mL, shown by shaded area) is shown. (**B**) IFN-γ levels in HD PBMCs (*n* = 5 HD) treated with exosomes (0.13 × 10^9^ particles/mL), isolated from supernatants of XRT- or PT-irradiated and non-irradiated (sham) HNSCC cells (Cal27, HNC208, HNC285, and HNC365), and activated with plate-bound anti-CD3 and anti-CD28 antibodies for 72 h. PBMCs not exposed to exosomes were used as controls. IFN-γ values from PBMCs not treated with exosomes were normalized as 1 for each experiment. Values are presented as mean ± SD, and symbols represent individual HDs. Significance was determined by one-way ANOVA (*p* < 0.0001 for each individual experiment shown) and post hoc testing was conducted using Tukey’s test. (**C**) Percent inhibition of IFN-γ shown in (**B**) compared to untreated (no exosome) PBMCs. Values are presented as mean ± SD, and symbols represent individual experiments (percent inhibition of IFN-γ levels in PBMCs exposed to exosomes from irradiated HNSCC cells, *n* = 4). Significance was determined by one-way ANOVA (*p* = 0.3481).

**Table 1 cancers-16-01008-t001:** Clinicopathologic features of individual HNSCC patients ^1^.

	HNC208	HNC285	HNC365
Age	77	72	56
Gender	Female	Male	Male
p16 status	Negative	Negative	Negative
Smoking	Non-smoker	Former, 75 pack years	Former, 19 pack years
Tumor grade	Well to moderately differentiated	Moderately differentiated	Moderately differentiated
Tumor stage	pT4aN0M0	pT4a pN2bcM0	pT4a pN3b

^1^ p16 negative indicates HPV negative. For smokers, pack years are an indicator of smoking status and calculated by multiplying the number of packs of cigarettes smoked per day by the number of years the individual has smoked. Tumor stage from T1 to T4 refers to tumor sizes. N1 to N3 refers to the involvement of regional lymph nodes depending on the number and location of the lymph nodes involved. N0 denotes the absence of cancer in the regional lymph nodes. M0 denotes the absence of metastasis. For patient HNC365, we do not know the metastatic status of the tumor.

## Data Availability

All data needed to evaluate the conclusions in the paper are presented in the paper and/or Appendix A. Additional data related to this study may be requested from the authors.

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
