# Peer review of "Proton Treatment Suppresses Exosome Production in Head and Neck Squamous Cell Carcinoma"

_cancers, 2024, doi:10.3390/cancers16051008_

Round 1
Reviewer 1 Report
Comments and Suggestions for Authors
This is an excellent study dealing with a promising emerging treatment modality (proton therapy) employed in head and neck squamous cell carcinomas. Investigators found that proton irradiation of primary cell cultures obtained from some surgically resected oral SCC HPV negative, compared to traditional photon therapy, reduced the production of exosomes limiting their suppressive effect on immune surveillance. These findings, if confirmed in larger series of HNSCC, could have significant therapeutic implications. Moreover, other subsites of head and neck carcinomas should be investigated in order to verify the real potential application of this novel cancer therapy.
Author Response
This is an excellent study dealing with a promising emerging treatment modality (proton therapy) employed in head and neck squamous cell carcinomas. Investigators found that proton irradiation of primary cell cultures obtained from some surgically resected oral SCC HPV negative, compared to traditional photon therapy, reduced the production of exosomes limiting their suppressive effect on immune surveillance. These findings, if confirmed in larger series of HNSCC, could have significant therapeutic implications. Moreover, other subsites of head and neck carcinomas should be investigated in order to verify the real potential application of this novel cancer therapy.
Thank you for your enthusiastic comments on our manuscript. We agree that in future studies we need to extend this research to other HNSCC subtypes. We have now included this statement in the Results and Discussion section (Lines 418-421).

Reviewer 2 Report
Comments and Suggestions for Authors
In this study the authors analyzed as proton treatment suppresses exosome production in head and 2 neck squamous cell carcinoma. The manuscript is very intestering and is written well.
In the Introduction section, the authors reported that “The current standard of care 45 is surgery followed by photon (X-ray)-based intensity-modulated radiation therapy 46 (IMRT), with or without chemotherapy”, but this sentence is incomplete.
In fact, as indicated in NCCN guidelines, the standard is surgery followed by RT with/out chemotherapy or definitive CRT or induction chemotherapy followed by concomitant CRT based on the stage of the disease and its anatomical location.
Recommended references:
- Guidelines Head and Neck Cancers Version: 2.2024 www.nccn.org
- https://doi.org/10.1002/hed.27332
The Authors report data from three patients with squamous cell tumors of the oral cavity, but do not mention if this sample is homogeneous or not in terms of characteristics (e.g. stage of the disease, tumoral grading, smoking, age, sex). Please specify this. In the Discussion section, the study limitations are missing. In particular, the sample is small and the patients analyzed are all 3 affected by an oral cavity cancer. This is not representative of all head and neck cancer (HNC) subtypes. Further studies including more patients with different types of HNC and their characteristics are needed.
Author Response
In this study the authors analyzed as proton treatment suppresses exosome production in head and 2 neck squamous cell carcinoma. The manuscript is very intestering and is written well.
We would like to thank the reviewer for the thoughtful remarks and constructive critiques. Below please find our responses to your comments.
In the Introduction section, the authors reported that “The current standard of care (line 45) is surgery followed by photon (X-ray)-based intensity-modulated radiation therapy (line 46) (IMRT), with or without chemotherapy”, but this sentence is incomplete. In fact, as indicated in NCCN guidelines, the standard is surgery followed by RT with/out chemotherapy or definitive CRT or induction chemotherapy followed by concomitant CRT based on the stage of the disease and its anatomical location.
Recommended references:
- Guidelines Head and Neck Cancers Version: 2.2024 www.nccn.org
- https://doi.org/10.1002/hed.27332
We have modified the line in the introduction (Line 45) and have made the necessary additions as suggested to accurately reflect the standard of care in HNC as per NCCN guidelines. Moreover, we added two new references (References 4 and 5) as suggested.
The Authors report data from three patients with squamous cell tumors of the oral cavity, but do not mention if this sample is homogeneous or not in terms of characteristics (e.g. stage of the disease, tumoral grading, smoking, age, sex). Please specify this.
We have now added a summary of the clinical and pathological features of the three oral cavity HNSCC patients in the Methods section as a new Table 1. (Line 113 and Lines 118-125) as requested. The patients were in the age range of 56-77 years and were quite similar in terms of tumor stage and tumor grade.
In the Discussion section, the study limitations are missing. In particular, the sample is small and the patients analyzed are all 3 affected by an oral cavity cancer. This is not representative of all head and neck cancer (HNC) subtypes. Further studies including more patients with different types of HNC and their characteristics are needed.
We agree with the reviewer’s comment and have now included their suggestion as a limitation of our study in the Results and Discussion section (Lines 418-421).
